# Structured follow-up pathway to support people after transient ischaemic attack and minor stroke (SUPPORT TIA): protocol for a feasibility study and process evaluation

Grace M Turner [ID] ,[1,2] Rachael Jones,[3,4] Phillip Collis,[1,2] Smitaa Patel,[5] Sue Jowett,[6] Sarah Tearne,[5] Robbie Foy [ID] ,[7] Lou Atkins,[8] Jonathan Mant,[9] Melanie Calvert [ID] [1,2,10,11,12,13]

For numbered affiliations see end of article.

**Correspondence to**
Dr Grace M Turner;
g.turner.1@bham.ac.uk

## ABSTRACT

**Introduction** People who experience transient ischaemic attack (TIA) and minor stroke have limited follow-up despite rapid specialist review in hospital. This means they often have unmet needs and feel abandoned following discharge. Care needs after TIA/minor stroke include information provision (diagnosis and stroke risk), stroke prevention (medication and lifestyle change) and holistic care (residual problems and return to work or usual activities). This protocol describes a feasibility study and process evaluation of an intervention to support people after TIA/minor stroke. The study aims to assess the feasibility and acceptability of (1) the intervention and (2) the trial procedures for a future randomised controlled trial of this intervention.

**Methods and analysis** This is a multicentre, randomised (1:1) feasibility study with a mixed-methods process evaluation. Sixty participants will be recruited from TIA clinics or stroke wards at three hospital sites (England). Intervention arm participants will be offered a nurse or allied health professional-led follow-up appointment 4 weeks after TIA/minor stroke. The multifaceted intervention includes: a needs checklist, action plan, resources to support management of needs, a general practitioner letter and training to deliver the intervention. Control arm participants will receive usual care. Follow-up will be self-completed questionnaires (12 weeks and 24 weeks) and a clinic appointment (24 weeks). Follow-up questionnaires will measure anxiety, depression, fatigue, health related quality of life, self-efficacy and medication adherence. The clinic appointment will collect body mass index, blood pressure, cholesterol and medication. Assessment of feasibility and acceptability will include quantitative process variables (such as recruitment and questionnaire response rates), structured observations of study processes, and interviews with a subsample of participants and clinical staff.

**Ethics and dissemination** Favourable ethical opinion was gained from the Wales Research Ethics Committee (REC) 1 (23 February 2021, REC reference: 21/WA/0036). Study results will be published in peer-reviewed journals and presented at conferences. A lay summary and

## Strengths and limitations of this study

⇒ The multicentre study will enable exploration of implementation of the intervention in the context of different sites.
⇒ The process evaluation is underpinned by the National Institutes of Health's Behavioural Change Consortium treatment fidelity framework.
⇒ Quantitative and qualitative methods will explore acceptability and how the intervention is implemented in practice.
⇒ Participants must have the ability to converse in everyday English and read in English to participate, which may limit the generalisability of our findings.

dissemination strategy will be codesigned with consumers. The lay summary and journal publication will be distributed on social media.

**Trial registration number** ISRCTN39864003.

## INTRODUCTION

Transient ischaemic attack (TIA) and minor stroke are important risk factors for stroke. Over 46 000 people experience a first TIA or minor stroke per year in the UK,[1] 240 000 in the USA[2] and 0.31 million in China.[3]

National guidelines promote long-term management that focuses on stroke prevention.[4–6] However, research shows TIA and minor stroke patients feel unsupported in stroke prevention—both medication and lifestyle change—and often lack basic understanding of their diagnosis, stroke risk and preventative medication.[7] Furthermore, many people experience a wide variety of residual impairments and unmet needs after TIA or minor stroke, including anxiety, mood/emotional impact, fatigue, cognitive impairment, physical weakness, visual impairment

**BMJ**

and impaired speech.[8–17] TIA and minor stroke have been also reported to impact on people's ability to return to work, performance at work, social activities and family relationships.[12–19] Follow-up care is variable and often inadequate with patients feeling abandoned after hospital discharge.[7]

Care needs after TIA and minor stroke include information provision (diagnosis and stroke risk); stroke prevention (medication and lifestyle change) and holistic care (residual problems and return to work or usual activities).[7] However, there is no evidence for how to best support these patients after rapid specialist review in hospital. To address this, we developed a multifaceted intervention which aims to actively identify and address unmet needs after TIA and minor stroke: Structured follow-Up Pathway to imProve management Of Residual impairmenTs and patients' quality of life after TIA and minor stroke. The components of the intervention are described in this protocol. In accordance with the Medical Research Council guidance on developing and evaluating complex interventions,[20] we will evaluate the feasibility and acceptability of (1) the intervention and (2) the trial procedures for a future randomised controlled trial (RCT) of this intervention. In addition, we will conduct a process evaluation to evaluate intervention fidelity and contextual influences on delivery.

## METHODS AND ANALYSIS
### Study design
The study is a multicentre, individual randomised feasibility study with a mixed-methods process evaluation. The study is reported in accordance with the Standard Protocol Items: Recommendations for Interventional Trials checklist[21] and the design is summarised in figure 1.

The study opened for recruitment on September 2021 with planned completion by December 2022.

### Patient and public involvement
A core group of three people who have experienced TIA or minor stroke have supported this study from inception,

with ad hoc contributions from other members of the public with TIA or minor stroke. The group supported the initial development of the research question and funding application, which were informed by their priorities and experiences. The group was involved in: selection of outcome measures; development of study documents; and design of the trial, such as recruitment strategies and considering participant burden related to data collection and attending intervention appointments. The group was integral to the intervention development, in particular the website of support services and resources. The group will continue to support the delivery of the study and dissemination of findings. One member (PC) is a coauthor and member of the study oversight committee. Patient and public involvement activities and impact will be reported using GRIPP2.[22]

### Study objectives
#### Trial design and methods
1. Assess feasibility and acceptability of the trial design and methods, including: number of patients meeting eligibility criteria; consent and randomisation processes; recruitment and retention rates; piloting the health economics questionnaire; and data completeness.
2. Provide data to inform the sample size for a definitive RCT.
3. Provide data to help inform selection of the primary outcome measure for a definitive RCT, including data completeness and correlation of the outcome measures with each other.

#### Intervention (process evaluation)
4. Investigate acceptability of the intervention for participants and intervention providers.
5. Test hypotheses relating to the theoretical underpinning of the intervention.
6. Assess if intervention providers are adequately trained to deliver the intervention.
7. Assess adherence to the intervention.
8. Assess contamination with the control group.
9. Define the 'dose' of the intervention (ie, attendance, length of appointment and number of appointments).
10. Explore how well intervention participants received and understood the intervention.
11. Explore to what extent the intervention was enacted as intended by patient participants (intervention group).

### Study setting and eligibility criteria
Participants will be recruited from TIA clinics and stroke wards at three tertiary hospital sites in England, one in South East England (Berkshire) and two in North West England (Wigan and Liverpool). Participants will be adults who have experienced a first or recurrent TIA or minor stroke. The full eligibility criteria are detailed in box 1.

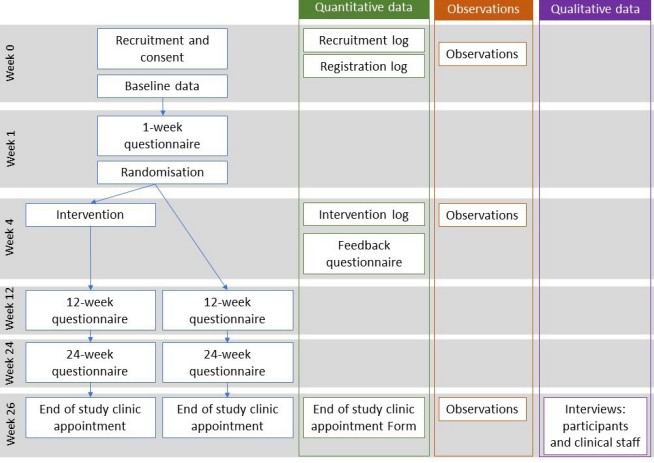

**Figure 1** Trial schema.

## Box 1 Eligibility criteria

### Inclusion criteria
⇒ Adults (aged ≥18 years).
⇒ Resident in England.
⇒ Diagnosis of confirmed transient ischaemic attack (TIA) or minor stroke by a stroke consultant. TIA will be defined as a transient episode of neurological dysfunction caused by focal brain, spinal cord or retinal ischaemia, without acute infarction.[30] Minor stroke will be defined as a modified Rankin scale score ≤1 or no change in modified Rankin scale score from pre-event (to account for people who were disabled prior to their TIA or minor stroke)*.
⇒ Attending the TIA clinic or stroke ward for a new diagnosis of TIA or minor stroke, rather than for a follow-up appointment.
⇒ Ability to converse in everyday English and read in English.
⇒ Capacity to provide fully informed consent for participation in the trial.

### Exclusion criteria
⇒ History of full stroke (modified Rankin scale score >1).
⇒ History of dementia.
⇒ People who lack capacity to participate, such as if they have severe memory problems that mean they would not remember giving consent or if they have severe communication problems, not precluding patients who use electronic devices to communicate.
⇒ Patients receiving early supported discharge or cardiac rehabilitation.
⇒ Patients receiving any palliative care.

* There is no standardised definition of minor stroke. Our criteria were selected as a practical definition to identify people with good functional recovery after stroke.[31]

## Intervention

Intervention development was underpinned by the Behaviour Change Wheel theoretical framework[23] and iteratively refined in collaboration with patient partners and a multidisciplinary team (online supplemental appendix 1, eTable 1).

The multifaceted intervention broadly comprises six components (figure 2):
1. Training for nurses and allied health professionals (AHPs) delivering the intervention.
2. Structured nurse or AHP led follow-up appointment, 4 weeks after TIA or minor stroke.
3. Needs checklist completed by participants prior to the appointment.

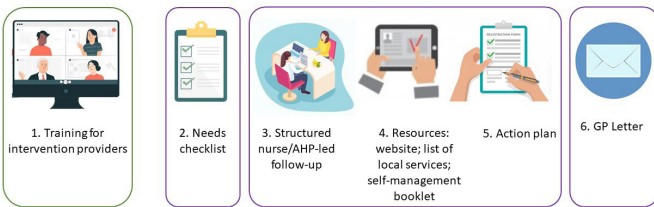

*This figure has been designed using resources from Freepik.com*

**Figure 2** Summary of the intervention components. AHP, allied health professional; GP, general practitioner.

4. Resources to support management of needs, including a website of resources and support services; list of local support services; and a self-management booklet.
5. Action plan.
6. Structured letter to general practitioners (GPs) to improve the interface communication between secondary and primary care.

Participants will also receive usual care and a Stroke Association TIA information sheet. Follow-up for TIA and minor stroke is not standardised; therefore, usual care varies between hospitals, GP practices and individual clinicians. Typically, any secondary care follow-up is related to imaging and investigations to determine cause of the TIA/ minor stroke and inform stroke risk prediction; for example, carotid imaging or ECG. Follow-up in primary care usually focuses on secondary prevention, such as medication and lifestyle advice; however, presence and quality of primary care follow-up post-TIA/minor stroke is variable.

Details of the intervention are described below in accordance with the template for intervention description and replication checklist.[24] The logic model is depicted in figure 3.

### Materials and procedures

Participants randomised to receive the intervention will be invited to a nurse/AHP-led follow-up appointment. Prior to their appointment, participants will be asked to complete a needs checklist, which will be posted to them prior to the appointment. The checklist comprises 12 potential needs which encompass information provision (diagnosis and stroke risk); secondary stroke prevention (medication and lifestyle change); and holistic care (psychological and psychosocial) (online supplemental appendix 2). The checklist is an adapted version of the Stroke Review Checklist[25] and was informed by the literature and earlier qualitative research,[7] codesigned with consumers.

The nurse/AHP will use the checklist to guide discussions to identify participants' unmet needs. If multiple needs are identified, priority will be given to addressing needs which the participant considers the most significant.

The nurse/AHP will address needs that can be resolved during the appointment, such as information about driving. For needs that cannot be immediately addressed, the nurse/AHP will, where appropriate, refer or signpost to support services and develop an action plan which will be agreed with the participant. Where possible, the nurse/AHP will make referrals; however, in some circumstances GP referral may be required, in which case this will be requested in the GP letter. To facilitate this, the intervention provider will be provided with a website of resources and support services and a list of local services.

The nurse/AHP will take the participants' blood pressure and, if raised (≥140/90 mm Hg), request for the participant's GP to review blood pressure in the action plan and GP letter.

| Intervention component | Mechanism of change | Outputs | Potential outcomes |
|---|---|---|---|
| 1. Training for nurses or AHPs delivering the intervention. | HCPs educated about potential needs after TIA or minor stroke, including health and social consequences. HCPs instructed how to identify and address potential unmet needs, including use of intervention materials. | ↑ HCPs knowledge of potential needs and strategies to identify and address needs. | *Short term* <br>• ↑ satisfaction with care. <br>• Access relevant further support (e.g. GP, support service). <br>• Access relevant resources (e.g. websites, apps). <br>• ↑ knowledge and understanding of diagnosis and stroke risk. |
| 2. Structured nurse or AHP led follow-up appointment, 4 weeks after TIA or minor stroke. | Service to provide patients with access to nurse or AHP follow-up. | Patients access holistic care and support for needs related to their TIA or minor stroke. Needs may be actioned immediately (e.g. driving information, reassurance) or an agreed plan for self-management or further support. | |
| 3. Needs checklist completed by participants prior to the appointment. | Checklist provides patients with the opportunity to reflect on their needs and facilitates communication with the HCP. | Patients' needs are actively identified and acknowledged by the HCP. | *Medium term* <br>• ↑ medication adherence. <br>• ↑ confidence and ability to self-manage needs. |
| 4. Resources to support management of needs, including a website of resources and support services; list of local support services; and a self-management booklet. | ↑ knowledge of relevant support services and resources. | Patients are referred or signposted to relevant support services. Patients are recommended relevant resources for their individual needs (e.g. websites, apps). | *Long term* <br>• ↑ health related quality of life. <br>• ↓ or improved residual problems (e.g. anxiety, fatigue). <br>• ↓ in stroke risk factors. |
| 5. Action plan. | Provides an opportunity for shared decision making and goal setting, and empowers patient to access services or resources and/or self-manage. | Patients are given clear actions to self-manage needs or access further support. | |
| 6. Structured GP letter | ↑ communication between secondary and primary care, and patients and GPs. Patients are empowered to access GP support. | GPs have better understanding of their patients' needs and care received. GPs receive clear recommendations for how to further support patient needs. | |

AHP: Allied Health Professional; GP: General Practitioner; HCP: Healthcare provider; TIA: Transient Ischaemic Attack; ↑: increase; ↓: decrease

**Figure 3** Logic model.

If necessary, the nurse/AHP may invite the participant to attend another follow-up appointment, at a suitable time point, to monitor the participant's progress and revise the action plan if required. These additional follow-ups may be conducted by telephone, video call or face to face.

A letter will be sent to the participant's GP along with a copy of the agreed action plan. A letter template will include recommended GP actions, a summary of the appointment and actions taken.

The participant will be provided with a self-management booklet (an abridged version of the resources and services website) and a copy of the action plan and GP letter.

### Intervention provider
The intervention will be delivered by a nurse or AHP, with stroke expertise, who are clinical staff at the participating hospital sites. It is anticipated that 1–2 intervention providers will be trained per site; however, this will depend on availability of clinical staff at sties. The nurses and AHPs will attend training which will include education about potential needs after TIA and minor stroke, and how to deliver the intervention. One training session, approximately 2.5 hours) will be provided remotely (via Zoom); however, ad hoc support and feedback will be encouraged after the training.

### Setting and modes of delivery
The intervention appointments will be delivered at the site's TIA clinic, either face to face or remotely (eg, telephone or video call). Face-to-face delivery will be preferable where possible.

### When and how often
The intervention appointment will take place at 4 weeks (or up to 6 weeks). The appointment is expected to last approximately 30 min. One appointment will be offered initially; however, participants will have an option to attend additional follow-up if judged clinically necessary by the nurse or AHP. There are no predetermined criteria for further follow-up and the criteria used by nurses/AHPs will be recorded as part of the feasibility study to inform future refinement of the intervention.

### Control arm
The control group will receive usual care and be given a Stroke Association TIA information sheet when they are informed about their allocation to the control arm.

### Recruitment
A member of the clinical team will screen patients' medical records and approach potentially eligible patients face to face or by phone. After confirming eligibility, potential participants will be invited to take part in the study. Informed consent may be taken face to face (for people approached in clinic), by post (for people who need more time to consider participation) or verbally (for people approached via phone). Verbal consent will be clearly documented in the participant's medical records and the participant will also be sent a postal consent form to compete. Sites will receive a per-participant reimbursement for recruitment.

### Sample size
The study will aim to recruit 60 participants (30 in the intervention group, 30 in the control group). As this is a feasibility study, no formal sample size calculation has

been performed; however, the sample size is the estimated number that would be feasible to show that we can recruit these types of patients for this type of study.[26]

## Randomisation

Participants will be randomised 1:1 to either the intervention or control group. A minimisation algorithm will be used within an online randomisation system to ensure balance in the treatment allocation using the following variables: age at consent (<60 years, ≥60 years); sex (male, female); diagnosis (TIA, minor stroke); employment (employed, non-employed/retired).

A 'random element' will be included in the minimisation algorithm, so that each patient has a probability, of being randomised to the opposite treatment that they would have otherwise received.

Participants will be randomised at baseline by clinical staff; however, to prevent baseline patient reported outcomes being affected by study arm allocation, participants will be notified of their randomisation allocation after they have returned the 1-week questionnaire or at 3 weeks (if the 1-week questionnaire is not returned). Participants will be notified of their allocation by a letter in the post, which will be sent by the research team at the Trials Unit. Due to the nature of the intervention, it is not possible to blind participants or clinicians delivering the intervention.

## Outcomes and data collection

Table 1 summarises the patient reported, health economic and clinical outcome measures. Contact details, demographic information and medical history will be collected at baseline from medical records or participant interview, by a member of clinical staff. Questionnaires comprising Patient-Reported Outcome Measures (PROMs) (table 2) will be completed by participants, either by post or electronically, at 1, 12 and 24 weeks. PROM rationale for assessment and psychometric properties are presented in online supplemental appendix, eTable 2. Questionnaires at 12 and 24 weeks will also collect health economics data. The first PROM completion will be at 1 week rather than baseline due to the nature of the PROM questions and to reduce burden on participants. Clinical data (table 2) will be collected at an end of study clinic appointment at 26 weeks by a research nurse or clinical staff. Where possible, this appointment will be face to face in the TIA clinic; however, may be delivered remotely if face to face is not an option.

### Feasibility outcomes and process evaluation

The feasibility study and process evaluation outcomes are detailed in tables 2 and 3. The process evaluation is underpinned by the National Institutes of Health's Behavioural Change Consortium treatment fidelity framework.[27] This framework includes five domains

**Table 1** Summary of patient reported, health economic and clinical outcome measures

|  | Data | Timepoint |
|---|---|---|
| Baseline data | Contact details | Baseline |
|  | Demographic: date of birth, sex, ethnicity, employment status |  |
|  | Medical: diagnosis, date of TIA or minor stroke, modified Rankin scale score, length of stay, smoking status, alcohol consumption, height, weight, body mass index, comorbidities, medication, blood pressure, cholesterol |  |
| Patient-reported outcome measure | Health related quality of life: Patient-Reported Outcomes Measurement Information System-Global Health 10 | 1, 12 and 24 weeks |
|  | Health related quality of life: 5-level EuroQol 5-Dimensions |  |
|  | Anxiety/depression: Hospital Anxiety and Depression Scale |  |
|  | Fatigue: Fatigue Assessment Scale |  |
|  | Self-efficacy: Patient Activation Measure-13 |  |
|  | Medication adherence: Medication Adherence Rating Scale–5 |  |
|  | Satisfaction with overall care after TIA/minor stroke question: 5-point Likert scale (very satisfied – very dissatisfied) |  |
| Health economics | Use of healthcare services | 12 and 24 weeks |
|  | Change in employment status, altered work hours and days off sick |  |
|  | Other costs incurred because of TIA or minor stroke |  |
| Clinical data | Body mass index | Baseline and 26 weeks |
|  | Blood pressure |  |
|  | Bloods: cholesterol |  |
|  | Medications |  |

TIA, transient ischaemic attack.

**Table 2** Feasibility outcomes and measurement of outcomes

| Objective | Feasibility outcomes | Measurement of outcome |
|---|---|---|
| (A) Assess feasibility and acceptability of the trial design and methods | No of eligible/ineligible patients and reasons for ineligibility | Recruitment log |
| | Proportion of participants who consent face to face, verbal or postal | Registration log: method of consent |
| | Willingness of clinical staff to randomise patients | Interviews (clinical staff involved in randomisation) |
| | Recruitment and attrition rates | Registration log |
| | Response rates and frequencies of missing data: participant completed questionnaires and case report forms | 1, 12 and 24 weeks questionnaires Case report forms |
| | End of study clinic appointment attendance rates | End of Study Clinic Appointment Form |
| | Acceptability of the trial design | Interviews (participants and clinical staff) Structured observations |
| (B) Provide data to inform the sample size for a definitive randomised controlled trial | SD of continuous patient reported outcome measures at 6 months | Patient reported outcome measure scores |
| | Recruitment and attrition rates | Registration log |
| (C) Provide data to help inform selection of the primary outcome measure for a definitive randomised controlled trial | Correlation of patient reported outcome measures | Patient reported outcome measure scores |
| | Patient reported outcome measure response rates and missing data | 1, 12 and 24 weeks questionnaires |

of treatment fidelity: Study Design, Training, Delivery, Receipt and Enactment.

### Case report forms

The following case report forms will collect data on feasibility outcomes: recruitment log (recruitment rates and reasons for ineligibility); registration log (method of consent: face to face/verbal/ postal); intervention log (attendance rates, duration, number of appointments per participant); end of study clinic appointment form (attendance). Case report forms will be assessed for missing data. The following intervention documents will capture information on needs, what was discussed and action plans: checklist, action plan and GP letter.

### Participant completed questionnaires

Participant completed questionnaires (1, 12 and 24 weeks) will be analysed for response rates and missing data. SDs of continuous PROMs at 6 months and correlation of PROMs will inform the sample size and selection of outcome measures for the definitive RCT. The intervention feedback questionnaire will report acceptability of the intervention. A paper copy of the feedback questionnaire and prepaid envelope will be posted to participants after the intervention appointment. This questionnaire contains 5-point Likert scale questions (eg, strongly agree—strongly disagree) and free text questions

about experiences of the checklist, appointment and action plan.

### Structured observations

A member of the study team will observe the following study processes: recruitment and consent procedures; intervention appointments; and end of study clinic appointments. Both face to face and remote modes of delivery will be observed for these procedures if possible. A target of three observations will be conducted for recruitment/consent and end of study clinic appointments (one at each site). A target of two intervention appointments will be observed per site (20%). More observations may be conducted if deemed necessary; for example, multiple clinical staff performing each procedure. A pragmatic approach will be taken to select which sessions to observe based on the availability of the research and clinical teams. A checklist will be used to document adherence to the protocol and field notes will be collected.

### Audit

At the end of the recruitment period, each site will perform an audit to identify the total number of confirmed TIA and minor stroke patients who attended the TIA clinic or stroke ward during the recruitment period. The age and sex of these patients will also be collected. This data

**Table 3** Process evaluations outcomes and measurement of outcomes

| NIH BCC domain | Objective | Outcome | Measurement of outcome |
|---|---|---|---|
| Study design | d) Investigate acceptability of the intervention for participants and intervention providers | Participants' and intervention providers' opinion on acceptability of the intervention | Interviews (participants and intervention providers) Feedback questionnaire (intervention participants) |
| | e) Test hypotheses relating to the theoretical underpinning of the intervention | Participants' satisfaction with identification and management of needs | Interviews (participants and intervention providers) Feedback questionnaire (intervention participants) |
| | | Participants acting on agreed action plans and/or accessing support services | Interviews (participants) |
| Training | f) Assess if intervention providers are adequately trained to deliver the intervention | Intervention providers' understanding of the intervention components | Interviews (intervention providers) |
| Delivery | g) Assess adherence to the intervention | Intervention providers' adherence to and deviations from the intervention manual | Structured observations Intervention log |
| | h) Assess contamination with the control group | Control group contamination | Interviews (participants and clinical staff) Structured observations |
| | i) Define the 'dose' of the intervention | Intervention follow-up appointment: attendance, length of appointment and number of appointments | Intervention log |
| Receipt | j) Explore how well intervention participants received and understood the intervention | Participants' perception of the intervention | Interviews (participants) Feedback questionnaire (intervention participants) |
| Enactment | k) Explore to what extent the intervention was enacted as intended by intervention participants | Participants acting on agreed action plans and/or accessing support services | Interviews (intervention participants) |

BCC, Behavioural Change Consortium; NIH, National Institutes of Health.

will be used to compare average age and sex of patients recruited to the trial against patients not recruited.

*Qualitative interviews*

At the end of the study, semistructured interviews will be conducted with a subset of participants and clinical staff involved in recruitment and/or intervention delivery. The sample size is anticipated to be 8–10 patients and 4–6 clinical staff (including those involved in recruitment/consent, intervention delivery and end of study clinic appointments). For patient participants, convenience sampling will be used initially; however, sampling will become increasingly purposeful to achieve variation in age (<60 years, ≥60 years) and diagnosis (TIA, minor stroke). For clinical staff, convenience sampling will be used. Interviews will be conducted by GT, an experienced qualitative researcher. Interviews will be face to face (home/ hospital), telephone or video call, depending on the participants preference. Interviews will explore acceptability of the intervention and trial design. Semistructured topic guides will include discussion of the following:

▶ Patient participants:

– Intervention: intervention and trial design acceptability; how well intervention participants received and understood the intervention; extent to which intervention providers addressed needs; if the action plan was actioned; facilitators and barriers to enactment.
– Control: trial design acceptability; intervention contamination.
– Both: what care/support participants received; understanding what comprised usual care.
▶ Staff participants: acceptability of the trial design; experience of training day and understanding of the intervention; acceptability of delivering the intervention; facilitators and barriers to implementing both the trial design and the intervention; and experience of contamination with the control group.

*Monitoring, adverse events and study oversight*

Information on trial monitoring, adverse events and study oversight is presented in online supplemental appendix 4.

**Analysis**

Quantitative outcomes will be analysed using simple descriptive statistics (eg, proportions and percentages,

**Table 4** Progression criteria

| Key uncertainties | Measures used | Progression criteria |
|---|---|---|
| **Trial design** | | |
| Recruitment | % target sample size recruited | ► ≥90%: proceed to a full-scale trial |
| | | ► 70%–89%: SOC will consider the feasibility of proceeding to a full-scale trial bearing in mind the data presented, representativeness of the sample and possible steps to increase recruitment. |
| | | ► <70%: full-scale trial unlikely to be feasible |
| Randomisation* | % of consented participants randomised | ► ≥90%: proceed to a full-scale trial |
| | | ► 70%–89%: SOC will consider the feasibility of proceeding to a full-scale trial bearing in mind the data presented, representativeness of the sample and possible steps to address randomisation issues. |
| | | ► <70%: full-scale trial unlikely to be feasible |
| Return rate of 24 weeks questionnaire* | % of 24 weeks questionnaires returned | ► ≥80%: proceed to a full-scale trial |
| | | ► 50%–79%: SOC will consider the feasibility of proceeding to a full-scale trial bearing in mind the data presented, representativeness of the sample and possible steps to increase return rates. |
| | | ► <50%: full-scale trial unlikely to be feasible |
| **Intervention** | | |
| Attendance rate* | % of intervention arm participants attending first appointment | ► ≥90%: proceed to a full-scale trial |
| | | ► 70%–89%: SOC will consider the feasibility of proceeding to a full-scale trial bearing in mind the data presented, representativeness of the sample and possible steps to increase attendance |
| | | ► <70%: full-scale trial unlikely to be feasible |
| Delivery of the intervention | % completion of: checklists, action plans, GP letters; use of directory of support services; Issues regarding delivery of the intervention components and contamination explored in qualitative interviews | The SOC will consider the quantitative and qualitative data and make an overall judgement on whether the intervention content is delivered as intended |
| Acceptability | % of participants reporting acceptability of intervention components on intervention feedback questionnaire; issues regarding acceptability of the intervention components explored in qualitative interviews | The SOC will consider the quantitative and qualitative data and make an overall judgement on whether the intervention is acceptable |

*Critical progression criteria: the trial is unlikely to be feasible if these criteria are not met, even if other criteria are satisfactory.
GP, general practitioner; SOC, Study Oversite Committee.

mean and SDs) and where appropriate, point estimates of effect sizes (eg, mean differences and relative risks) and associated 95% CIs. Analyses comparing the intervention and control groups will use the intention-to-treat principle, that is, all participants will be analysed in the treatment group to which they were randomised irrespective of compliance or other protocol deviation. Analysis will be conducted using Stata V.16.

For qualitative data, interviews will be audiorecorded and transcribed verbatim. NVivo V.12 will be used to manage, sort, code and organise the anonymised transcribed data. Interview transcripts will be analysed by GT using directed thematic analysis, using Braun and Clarke's six-stage process,[28] informed by the research aims.[29]

The health economics analysis will assess completion rates, estimate resources required to deliver the intervention and report simple descriptive statistics for resource use and outcomes. Key resource use items not currently specified on the form but included by participants will also be identified. The information will inform the cost and outcome data collection and identification of unit costs for a larger trial.

As this project is a training fellowship, the fellow (GT) will conduct the analysis and will have access to the whole dataset in order to conduct the trial. Therefore, it is not possible to conducted blinded analyses.

Data will be made available on reasonable request.

## Progression criteria

The predefined progression criteria, detailed in table 4, will be used to inform a decision on whether a full RCT is warranted and feasible. The criteria were agreed by the Study Oversight Committee and follow a traffic light system using quantitative measures supported by qualitative data.

## ETHICS AND DISSEMINATION

Favourable ethical opinion was gained from the Wales Research Ethics Committee (REC) 1 (23 February 2021, REC reference: 21/WA/0036). Study results will be published in a peer-reviewed journal and presented at relevant conferences. A lay summary and dissemination strategy will be codesigned with consumers. The lay summary and peer review publication will be distributed on social media.

**Author affiliations**
[1]Institute of Applied Health Research, University of Birmingham, Birmingham, UK
[2]Centre for Patient Reported Outcomes Research and Institute of Applied Health Research, University of Birmingham, Birmingham, UK
[3]Royal Wolverhampton NHS Trust, Wolverhampton, UK
[4]Clinical Research Network West Midlands, West Midlands, UK
[5]Birmingham Clinical Trials Unit, University of Birmingham, Birmingham, UK
[6]Health Economics Unit, University of Birmingham, Birmingham, UK
[7]Leeds Institute of Health Sciences, University of Leeds, Leeds, UK
[8]Centre for Behaviour Change, University College London, London, UK
[9]Department of Public Health and Primary Care, University of Cambridge, Cambridge, UK
[10]NIHR Surgical Reconstruction and Microbiology Research Centre, University Hospitals Birmingham NHS Foundation Trust and University of Birmingham, Birmingham, UK
[11]NIHR Birmingham Biomedical Research Centre, University Hospitals Birmingham NHS Foundation Trust and University of Birmingham, Birmingham, UK
[12]Birmingham Health Partners Centre for Regulatory Science and Innovation, University of Birmingham, Birmingham, UK
[13]NIHR Applied Research Collaboration (ARC) West Midlands, University of Birmingham, Birmingham, UK

**Acknowledgements** We thank the patient and public involvement group members who have provided valuable insight and feedback into the intervention and study design: Phil Collis, Lesley Thomson and Sally Hughes. We thank those involved in the development of the intervention with special mention to Kirsty Price and Joanne Chesson.

**Contributors** GT, MC, JM, RF and LA contributed to the conception of the study. The design of the feasibility study was led by GT with contributions from MC, JM, RF, LA, PC, SP, SJ and ST. The design of the intervention was led by GT with contributions from MC, JM, RF, LA, PC and RJ. GT wrote the first draft of the manuscript. All authors contributed to and approved the final version.

**Funding** This work was supported by National Institute for Health Research (NIHR) Post-Doctoral Fellowship Scheme grant number PDF-2017-10-047.

**Disclaimer** The views expressed in this publication are those of the author(s) and not necessarily those of the NIHR or the Department of Health and Social Care.

**Competing interests** MC and JM are National Institute for Health Research (NIHR) Senior Investigators. MC also receives funding from the National Institute for Health Research (NIHR), UK Research and Innovation (UKRI), the NIHR Birmingham Biomedical Research Centre, NIHR Surgical Reconstruction and Microbiology Research Centre and NIHR ARC West Midlands at the University of Birmingham and University Hospitals Birmingham NHS Foundation Trust, UK SPINR (UKRI) Health Data Research UK, Innovate UK (part of UKRI), Macmillan Cancer Support, UCB Pharma, GSK, Gilead and Janssen. MC has received personal fees from Astellas, Aparito, CIA Oncology, Takeda, Merck, Daiichi Sankyo, Glaukos, GSK and the Patient-Centered Outcomes Research Institute (PCORI) outside the submitted work. JM has done consultancy work for BMS/Pfizer, Omron and Doctorlink Innovations.

**Patient and public involvement** Patients and/or the public were involved in the design, or conduct, or reporting, or dissemination plans of this research. Refer to the Methods section for further details.

**Patient consent for publication** Not applicable.

**Provenance and peer review** Not commissioned; externally peer reviewed.

**ORCID iDs**
Grace M Turner http://orcid.org/0000-0002-9783-9413
Robbie Foy http://orcid.org/0000-0003-0605-7713
Melanie Calvert http://orcid.org/0000-0002-1856-837X

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
