## [Reviewer comments · BMJ Open]

ARTICLE DETAILS

TITLE (PROVISIONAL)	Structured follow-up pathway to support people after transient ischaemic attack and minor stroke (SUPPORT TIA): Protocol for a feasibility study and process evaluation
AUTHORS	Turner, Grace; Jones, Rachael; Collis, Phillip; Patel, Smitaa; Jowett, Sue; Tearne, Sarah; Foy, Robbie; Atkins, Lou; Mant, Jonathan; Calvert, Melanie

VERSION 1 – REVIEW

REVIEWER	Hillier, Susan University of South Australia, Sansom Institute for Health Research
REVIEW RETURNED	27-Jan-2022

GENERAL COMMENTS	Thank you for the opportunity to read this protocol. Could you please just clarify that this is to INFORM a future definitive RCT? sometimes it reads like you are running the definitive RCT in parallel and sometimes like this feasibility trial will run first followed by a full RCT. Maybe a simple timeline to illustrate?
---

REVIEWER	Crowfoot, Gary University of Newcastle, School of Health Sciences
REVIEW RETURNED	15-Feb-2022

GENERAL COMMENTS	Thank you for the opportunity to review this protocol. This feasibility study and process evaluation outlines a robust approach to exploring this intervention to address care issues for people with minor stroke and TIA. There are several minor amendments that should be made and some considerations for the authors to review: Amendments 1. The opening sentence of your abstract is convoluted and should be separated into 2 sentences. E.g. "People who experience Transient Ischaemic Attack (TIA) and minor stroke have limited follow up despite rapid specialist review in hospital. This means they often have unmet needs and feel abandoned following discharge"2. Usual care should be clarified for context. Does usual care mean that they are provided with a referral to see their GP on discharge or are other cares provided?3. It is unclear how you are able to measure or determine the dose of the intervention in the process evaluation. This design does not use a dose-escalation framework and the intervention cohorts appear to receive a static dose. Please clarify what is meant here (study objectives - pg 6)
--

	4. A methodology for the analysis of qualitative interview data has not been provided. Please outline what methodology or approach the authors will use to complete the qualitative data analysis. E.g. Braun & Clarke's (2006) qualitative descriptive approach Considerations:  1. Terminology varies widely across the world for consumer involvement in research. You have used the term patient partners in this protocol. However, this terminology could demonstrate a power imbalance (connotations of the medicalised term "patient"). Please consider calling them consumers 2. Consider revising the format of table 1 as the potential outcomes column is merged for all rows. It was unclear that your potential outcomes apply to all rows on my first review of the table. The use of a margin line may assist with clarity. 3. Stress is likely an important outcome to measure that would be of interest to measure in this study. This could be readily measured by using the Depression, Anxiety & Stress Scale (DASS-21) instead of the HADS as proposed.
--	--

REVIEWER	Finch, Emma University of Queensland
REVIEW RETURNED	24-Feb-2022

GENERAL COMMENTS	Thank-you for the opportunity to review this protocol for a study trialling a new follow-up pathway to support people after TIA and minor stroke. Overall, the protocol is well-written and address an important area of need internationally. Specific comments:  - Could the literature review be expanded to provide more context and information about the consequences of TIA and minor stroke? As BMJ Open is an international journal, could figures for other regions be provided too? - The study is well-planned and has a strong theoretical basis. - The patient and public involvement behind the study is strong. - The logic model is comprehensive and easy to follow. - The inclusion of progression criteria is a strength. Best wishes to the authors for their research and I look forward to reading the study results in the future.
--

REVIEWER	Purvis, Tara Monash University, Stroke and Ageing, School of Clinical Sciences at Monash Health
REVIEW RETURNED	27-Feb-2022

GENERAL COMMENTS	Thank you for this protocol paper outlining a multicentre, randomised, feasibility study and process evaluation of an intervention to support people after transient ischaemic attack and minor stroke (SUPPORT TIA), which will be used to inform a future definitive randomised trial. The paper is well written, and this is an important area of research, however, there are several areas that could be improved, particularly around the detail included to allow for replication and transparency  - The aims stated in the abstract and main text are to assess the 'feasibility and acceptability of i) the intervention and ii) a RCT of this intervention'. I am not 100% clear on the difference – does ii)
---

refer more to the trial procedures? I wonder if the aims could be more clearly stated for the reader

Abstract

-Within the abstract, outcomes within the 12 & 24-week questionnaires are mentioned, but it is unclear what is being measured at the 24-week clinic appointment. It would also be helpful to explain what 'quantitative process variables' refer to in the abstract

-There is also a switch between site staff (abstract) and clinical staff, (also including intervention provider) which is a little confusing for the reader. Keeping to the same terms, and clearly delineating any differences in clinical staff involved in the recruitment/randomisation and provision of the intervention would be helpful

Main text

- For context, it would be useful to give an overview of the 3 recruiting hospitals, i.e. ? tertiary hospitals, are they located in the same area etc

- Is pre-morbid function (or comorbidities) a consideration for inclusion (other than memory/communication)? Are people living in residential care eligible?

- Please clarify the exclusion criteria of 'full stroke'

- Although this is only a feasibility study, and power calculations are not required, it would still be useful to understand how was the 30 participants in each arm determined? Was this based on prior work in the area, or determined by time frames, funding?

- What does usual care for patients with minor stroke and TIA entail in the UK – I think this is important for an international audience to have an understanding of this, and be more transparent on what the control group will receive

- What measures will be used to assess for contamination with the control group, and how do you anticipate this may occur? If patients in the control group receive input from other health professionals outside of trial, is this information captured. Similarly, are specifics related to what was accessed by intervention participants recorded?

-To assist with flow, it might be helpful if the authors consider including information about the 'intervention provider', 'setting and modes of delivery' & 'when & how often' earlier in the 'materials and procedures' section.

- Are hospital/clinical staff responsible for randomisation? Is there a stipend or reimbursement/payments for recruitment/randomisation of participants?

- Considering the literature in the area, it seems likely that multiple needs will be identified by participants. Priority will be given to how many areas i.e is there a limit? Is there capacity for the participant to add/alter their identified needs, as these will likely change over time?

- Please clarify what criteria will be used to determine if another follow-up appointment will be necessary, and after how long? This needs to be standardised across all nurse/AHP delivering the intervention

- Is there any further F/U on strategies put in place in the action plan – or is support then outsourced? Will nurse/AH (intervention deliverer) make referrals or is this the GP role?

- Additional information about the specific training of the intervention provider i.e the nurses/AHP delivering the intervention is required. How many staff do you anticipate having to train to

	deliver the intervention to 30 participants i.e will it be the same few delivering the intervention? Are they from the hospital sites involved in recruitment, or external? How many training sessions will be delivered, of approx. how long, will the training be face-to-face, or via video etc. As this is an important aspect to the intervention, is the training being evaluated?  - Please review what table (?should be 2 rather than 1) is being referred to in first paragraph of 'Definitive RCT' - Additional information about the collection of outcome data is required. For example, how is the baseline questionnaire being administered and by whom, and how is the needs checklist delivered to the participant? In addition, who is conducting the end of study clinic appointment, and how/where will this occur i.e face-to-face, clinic/home, video? Please also confirm timeframe for the end of clinic appointment – in figure 1 it seems this occurs at 26 weeks, but in table 2 it is 24 weeks? Outcomes:  -Is specific information related to what is discussed and actioned in the follow-up meeting obtained, as it would likely be important in also describing the 'dose'? Is this collected within the intervention log? -Is any feedback being obtained from GPs related to the structured letter? It would seem that for the definitive RCT, it would be relevant and important to obtain feedback related to how useful this is, and how it fits in with their current processes? - For the satisfaction with care question – does this relate to overall care after TIA/minor stroke, or specifically after the follow-up intervention. It appears to be included in the 1, 12 and 24 questionnaire? - The 'feedback questionnaire' for intervention participants is mentioned in Table 4, and briefly in the section on 'participant completed questionnaires' – but there is no information of how and when this is administered, and the specifics of what is included i.e. open vs closed questions, based on a framework – this needs further explanation - For the structured observations, it is noted that approx. 1 observation session for the recruitment/consent, intervention delivery and end of study clinical appointment will be conducted from each site – will there be multiple staff from each site performing each task? And therefore, more than 1 observation may be required? It would seem that observations of the intervention delivery component is quite important for fidelity – have the authors considered more observations of this aspect? - Authors state that 4-6 clinical staff are likely to be interviewed. Does this include both those involved in the recruitment/randomisation processes as well as the intervention delivery? As there is a mixture of clinical staff/intervention providers it is not 100% clear - Additional information specifically related to the interviews should be included. For example, who will conduct/analyse these i.e a study staff member not involved in the study delivery? Are the interviews to be conducted face-to-face, or via phone, video? Will they be audio recorded, transcribed verbatim? Will an interview guide be used? What broad topics are to be covered? Analysis:  - Is the planned analysis intention to treat? - Please define all ACRONYMS in tables/figures i.e. eTable 1 PROMs used e.g. HADS, PAM-123 - What is the current trial status?
--	---

	- Authors have included the SPIRIT guideline in the submission. If this has been used as a reporting guideline, then it would seem wise to refer to and cite in the protocol paper? Within the SPIRIT checklist, it is not clear what REC protocol refers to in relation to the submitted paper. Inclusion of these items in the paper is recommended ie. is there a process in place if participants want to withdraw, how are adverse events collected, what trial governance and data monitoring is in place?
--	---

VERSION 1 – AUTHOR RESPONSE

Reviewer: 1

Prof. Susan Hillier, University of South Australia Comments to the Author:

Thank you for the opportunity to read this protocol.

Could you please just clarify that this is to INFORM a future definitive RCT? sometimes it reads like you are running the definitive RCT in parallel and sometimes like this feasibility trial will run first followed by a full RCT. Maybe a simple timeline to illustrate?

To clarify, this protocol is for a feasibility study to inform a future definitive RCT. To avoid confusion, we have removed reference to a definitive RCT in the 'outcomes and data collection' section (page 12, lines 32 and 34) and Table 2. Furthermore, we have changed the subheading from "feasibility study and process evaluation" to "feasibility *outcomes* and process evaluation" (page 12, line 6). The definitive RCT is not yet funded; therefore, we have not included a timeline as suggested.

Reviewer: 2

Dr. Gary Crowfoot, University of Newcastle, University of Newcastle Hunter Medical Research Institute Comments to the Author:

Thank you for the opportunity to review this protocol. This feasibility study and process evaluation outlines a robust approach to exploring this intervention to address care issues for people with minor stroke and TIA. There are several minor amendments that should be made and some considerations for the authors to review:

Amendments

1. The opening sentence of your abstract is convoluted and should be separated into 2 sentences. E.g. "People who experience Transient Ischaemic Attack (TIA) and minor stroke have limited follow up despite rapid specialist review in hospital. This means they often have unmet needs and feel abandoned following discharge"

We have amended the sentence as suggested, page 3 lines 3-5.

2. Usual care should be clarified for context. Does usual care mean that they are provided with a referral to see their GP on discharge or are other cares provided?

We have added information on usual care on page 8, lines 1-7:

"Follow-up for TIA and minor stroke is not standardised; therefore, usual care varies between hospitals, GP practices and individual clinicians. Typically, any secondary care follow-up is related to imaging and investigations to determine cause of the TIA/ minor stroke and inform stroke risk prediction; for example, carotid imaging or electrocardiogram. Follow-up in primary

care usually focuses on secondary prevention, such as medication and lifestyle advice; however, presence and quality of primary care follow-up post-TIA/ minor stroke is variable.”

3. It is unclear how you are able to measure or determine the dose of the intervention in the process evaluation. This design does not use a dose-escalation framework and the intervention cohorts appear to receive a static dose. Please clarify what is meant here (study objectives - pg 6)

We have used the term “dose” to align with the terminology used in the National Institutes of Health’s (NIH) Behavioural Change Consortium (BCC) treatment fidelity framework. In the context of this feasibility study “dose” of the intervention is attendance, length of appointment and number of appointments. We have clarified this in the study objectives (page 6, lines 18-19).

“i) Define the “dose” of the intervention (i.e. attendance, length of appointment and number of appointments);”

4. A methodology for the analysis of qualitative interview data has not been provided. Please outline what methodology or approach the authors will use to complete the qualitative data analysis. E.g. Braun & Clarke's (2006) qualitative descriptive approach

We have added this information to page 16, lines 1-2:

“Interview transcripts will be analysed by GT using directed thematic analysis, using Braun and Clarke’s 6-stage process,(28) informed by the research aims.”

Considerations:

1. Terminology varies widely across the world for consumer involvement in research. You have used the term patient partners in this protocol. However, this terminology could demonstrate a power imbalance (connotations of the medicalised term "patient"). Please consider calling them consumers

Thank you for this suggestion, we have changed the term patient partners to consumers throughout the manuscript.

2. Consider revising the format of table 1 as the potential outcomes column is merged for all rows. It was unclear that your potential outcomes apply to all rows on my first review of the table. The use of a margin line may assist with clarity.

Thank you for this suggestion, we have revised the formatting of Table 1 for clarity.

3. Stress is likely an important outcome to measure that would be of interest to measure in this study. This could be readily measured by using the Depression, Anxiety & Stress Scale (DASS-21) instead of the HADS as proposed.

We agree that stress would be of interest to measure. However, the HADS was selected because it is widely used in TIA/ minor stroke research and its comprehensive psychometric evaluation. Furthermore, HADS was favoured by our lived experience advisory group.

Reviewer: 3

Dr. Emma Finch, University of Queensland Comments to the Author:

Thank-you for the opportunity to review this protocol for a study trialling a new follow-up pathway to support people after TIA and minor stroke. Overall, the protocol is well-written and address an important area of need internationally.

Specific comments:

- Could the literature review be expanded to provide more context and information about the consequences of TIA and minor stroke?

We have added additional information about consequences of TIA and minor stroke (page 5, lines 8-13):

“Furthermore, many people experience a wide variety of residual impairments and unmet needs after TIA or minor stroke, including anxiety, mood/ emotional impact, fatigue, cognitive impairment, physical weakness, visual impairment and impaired speech.(8-17) TIA and minor stroke have been also reported to impact on people’s ability to return to work, performance at work, social activities and family relationships.(12-19)”

As BMJ Open is an international journal, could figures for other regions be provided too?

We have now added figures for the USA and China (page 5, lines 3-4):

“Over 46,000 people experience a first TIA or minor stroke per year in the United Kingdom(1), 240,000 in the United States(2) and 0.31 million in China(3).”

- The study is well-planned and has a strong theoretical basis.**
- The patient and public involvement behind the study is strong.**
- The logic model is comprehensive and easy to follow.**
- The inclusion of progression criteria is a strength.**

Best wishes to the authors for their research and I look forward to reading the study results in the future.

Thank you for these positive comments.

Reviewer: 4

Ms. Tara Purvis, Monash University

Comments to the Author:

Thank you for this protocol paper outlining a multicentre, randomised, feasibility study and process evaluation of an intervention to support people after transient ischaemic attack and minor stroke (SUPPORT TIA), which will be used to inform a future definitive randomised trial.

The paper is well written, and this is an important area of research, however, there are several areas that could be improved, particularly around the detail included to allow for replication and transparency

- The aims stated in the abstract and main text are to assess the ‘feasibility and acceptability of i) the intervention and ii) a RCT of this intervention’. I am not 100% clear on the difference – does ii) refer more to the trial procedures? I wonder if the aims could be more clearly stated for the reader

Yes, (ii) refers to the trial procedures, we have re-worded the aim to clarify this in the abstract and main text (page 5, line 24):

“The study aims to assess the feasibility and acceptability of (i) the intervention and (ii) *the trial procedures* for a *future* randomised controlled trial of this intervention.”

Abstract

-Within the abstract, outcomes within the 12 & 24-week questionnaires are mentioned, but it is unclear what is being measured at the 24-week clinic appointment. It would also be helpful to explain what ‘quantitative process variables’ refer to in the abstract

We have added information what is being measured at the clinic appointment to the abstract (page 3, lines 21-22):

“The clinic appointment will collect body mass index, blood pressure, cholesterol and medication.”

We have also added examples of quantitative process variables (page 3, line 23):

“Assessment of feasibility and acceptability will include quantitative process variables *(such as recruitment and questionnaire response rates)*, ...”

-There is also a switch between site staff (abstract) and clinical staff, (also including intervention provider) which is a little confusing for the reader. Keeping to the same terms, and clearly delineating any differences in clinical staff involved in the recruitment/randomisation and provision of the intervention would be helpful

We have amended the abstract to use the term “clinical staff” for consistency with the manuscript.

We have also defined differences in clinical staff involved in the recruitment/randomisation and provision of the intervention in Tables 3 and 4.

Main text

- For context, it would be useful to give an overview of the 3 recruiting hospitals, i.e. ? tertiary hospitals, are they located in the same area etc

We have added an overview of the 3 hospitals (page 6, lines 25-26):

“Participants will be recruited from TIA clinics and stroke wards at three *tertiary* hospital sites in England, *one in South East England (Berkshire) and two in North West England (Wigan and Liverpool)*.”

- Is pre-morbid function (or comorbidities) a consideration for inclusion (other than memory/communication)? Are people living in residential care eligible?

There are no additional considerations for inclusion or exclusion other than those details in Box 1.

- Please clarify the exclusion criteria of 'full stroke'

We have clarified the definition of "full stroke" in box 1:

"History of full stroke (modified Rankin scale score >1)"

- Although this is only a feasibility study, and power calculations are not required, it would still be useful to understand how was the 30 participants in each arm determined? Was this based on prior work in the area, or determined by time frames, funding?

We have added further information to page 11, lines 16-18.

"As this is a feasibility study, no formal sample size calculation has been performed; **however, the sample size is the estimated number that would be feasible to show that we can recruit these types of patients for this type of study.**"

Reference: Teare MD, Dimairo M, Shephard N, et al. Sample size requirements to estimate key design parameters from external pilot randomised controlled trials: a simulation study. *Trials*. 2014;15(1):264.

- What does usual care for patients with minor stroke and TIA entail in the UK – I think this is important for an international audience to have an understanding of this, and be more transparent on what the control group will receive

We have added information on usual care on page 8, lines 1-7:

"Follow-up for TIA and minor stroke is not standardised; therefore, usual care varies between hospitals, GP practices and individual clinicians. Typically, any secondary care follow-up is related to imaging and investigations to determine cause of the TIA/ minor stroke and inform stroke risk prediction; for example, carotid imaging or electrocardiogram. Follow-up in primary care usually focuses on secondary prevention, such as medication and lifestyle advice; however, presence and quality of primary care follow-up post-TIA/ minor stroke is variable."

- What measures will be used to assess for contamination with the control group, and how do you anticipate this may occur? If patients in the control group receive input from other health professionals outside of trial, is this information captured. Similarly, are specifics related to what was accessed by intervention participants recorded?

Contamination will be explored through qualitative interviews with control participants and clinical staff (intervention provides and those involved in recruitment and trial processes). Control participant qualitative interviews will also explore if this group receive input from healthcare providers outside of the trial, e.g. GPs. Similarly, intervention participant qualitative interviews will explore what care or support these participants accessed.

We have added this information to the manuscript for clarity (page 15, lines 23-34):

"Semi-structured topic guides will include discussion of the following:

- Patient participants:
 - Intervention: intervention and trial design acceptability; how well intervention participants received and understood the intervention; extent to which intervention providers addressed needs; facilitators and barriers to enactment.
 - Control: trial design acceptability; intervention contamination.
 - Both: what care/support participants received; understanding what comprised usual care.

- Staff participants: acceptability of the trial design; experience of training day and understanding of the intervention; acceptability of delivering the intervention; facilitators and barriers to implementing both the trial design and the intervention; and experience of contamination with the control group.”

-To assist with flow, it might be helpful if the authors consider including information about the ‘intervention provider’, ‘setting and modes of delivery’ & ‘when & how often’ earlier in the ‘materials and procedures’ section.

Following your suggestion, we have moved the “Intervention” section to before the “Recruitment”, “Sample size” and “Randomisation” sections.

- Are hospital/clinical staff responsible for randomisation? Is there a stipend or reimbursement/payments for recruitment/randomisation of participants?

Randomisation is complete by clinical staff, we have clarified this on page 11, line 14.

“Participants will be randomised at baseline **by clinical staff**...”

Sites will receive a per-participant payment for recruitment, we have now detailed this on page 11, lines 1-2:

“Sites will receive a per-participant reimbursement for recruitment.”

- Considering the literature in the area, it seems likely that multiple needs will be identified by participants. Priority will be given to how many areas i.e is there a limit? Is there capacity for the participant to add/alter their identified needs, as these will likely change over time?

As detailed in the manuscript, if multiple needs are identified, priority will be given to addressing needs which the participant considers the most significant (page 10, line 12) and there is an option for additional follow-up if required (page 10, lines 22-24). Therefore, there is no pre-specified limit and the additional follow-up (if required) will provide an opportunity for participants to add/ alter their needs.

- Please clarify what criteria will be used to determine if another follow-up appointment will be necessary, and after how long? This needs to be standardised across all nurse/AHP delivering the intervention

As detailed in the manuscript, additional follow-up will be based on clinical judgment by the nurse or AHP. There are no pre-determined criteria for further follow-up and the criteria used by nurses/AHPs will be recorded as part of the feasibility study to inform future refinement of the intervention. We have clarified this in the manuscript, page 11, lines 1-2:

“There are no pre-determined criteria for further follow-up and the criteria used by nurses/AHPs will be recorded as part of the feasibility study to inform future refinement of the intervention.”

- Is there any further F/U on strategies put in place in the action plan – or is support then outsourced? Will nurse/AH (intervention deliverer) make referrals or is this the GP role?

There will be no planned follow-up of the action plan. The scope of support provided by nurses/ AHPs for the intervention is limited to that detailed in the manuscript. As part of the feasibility outcomes and process evaluation, qualitative interviews will explore if participants actioned their action plan or received further care/ support beyond the intervention (page 15, lines 24-26).

In terms of referrals, where possible intervention providers will make referrals; however, there will be some circumstances where only GPs can refer, in which case this will be requested in the GP letter. We have added this information to page 10, line 16-17:

“Where possible, the nurse/AHP will make referrals; however, in some circumstances GP referral may be required, in which case this will be requested in the GP letter.”

- Additional information about the specific training of the intervention provider i.e the nurses/AHP delivering the intervention is required. How many staff do you anticipate having to train to deliver the intervention to 30 participants i.e will it be the same few delivering the intervention? Are they from the hospital sites involved in recruitment, or external? How many training sessions will be delivered, of approx. how long, will the training be face-to-face, or via video etc. As this is an important aspect to the intervention, is the training being evaluated?

Thank you for this suggestion, we have added more information to page 10, lines 30-35.

“The intervention will be delivered by a nurse or AHP, with stroke expertise, who are clinical staff at the participating hospital sites. It is anticipated that 1-2 intervention providers will be trained per site; however, this will depend on availability of clinical staff at sites. The nurses and AHPs will attend training which will include education about potential needs after TIA and minor stroke, and how to deliver the intervention. One training session, approximately 2.5 hours) will be provided remotely (via Zoom); however, ad hoc support and feedback will be encouraged after the training.”

Experience of training day and understanding of the intervention will be explored in qualitative interviews with intervention providers (page 15, lines 31-34).

- Please review what table (?should be 2 rather than 1) is being referred to in first paragraph of ‘Definitive RCT’

Thank you highlighting, this typo has been amended (page 11, line 33).

- Additional information about the collection of outcome data is required. For example, how is the baseline questionnaire being administered and by whom, and how is the needs checklist delivered to the participant? In addition, who is conducting the end of study clinic appointment, and how/where will this occur i.e face-to-face, clinic/home, video? Please also confirm timeframe for the end of clinic appointment – in figure 1 it seems this occurs at 26 weeks, but in table 2 it is 24 weeks?

Thank you for this suggestion, we have added further information as requested:

Baseline data, page 11, line 35:

“Contact details, demographic information and medical history will be collected at baseline **from medical records or participant interview, by a member of clinical staff.**”

Needs checklist, page 10, line 4:

“Prior to their appointment, participants will be asked to complete a needs checklist, **which will be posted to them prior to the appointment.**”

End of study clinic appointment: timeframe edited to 26 weeks in table 2 and further detail added to page 11, line 42:

“Clinical data (Table 2) will be collected at an end of study clinic appointment at 26 weeks **by a research nurse or clinical staff. Where possible, this appointment will be face-to-face in the TIA clinic; however, may be delivered remotely if face-to-face is not an option.”**

Outcomes:

-Is specific information related to what is discussed and actioned in the follow-up meeting obtained, as it would likely be important in also describing the ‘dose’? Is this collected within the intervention log?

This information will be captured in the checklist, intervention log, action plan, GP letter. We have clarified this on page 14, lines 7-8:

“The following intervention documents will capture information on needs, what was discussed and action plans: checklist, action plan and GP letter.”

-Is any feedback being obtained from GPs related to the structured letter? It would seem that for the definitive RCT, it would be relevant and important to obtain feedback related to how useful this is, and how it fits in with their current processes?

Feedback will not be obtained from GPs due to time and resource constraints.

- For the satisfaction with care question – does this relate to overall care after TIA/minor stroke, or specifically after the follow-up intervention. It appears to be included in the 1, 12 and 24 questionnaire?

For clarity we have added more information about this question to table 2:

“Satisfaction with overall care after TIA/minor stroke question: 5-point Likert scale (very satisfied – very dissatisfied)”

- The ‘feedback questionnaire’ for intervention participants is mentioned in Table 4, and briefly in the section on ‘participant completed questionnaires’ – but there is no information of how and when this is administered, and the specifics of what is included ie. open vs closed questions, based on a framework – this needs further explanation

We have added more information about the feedback questionnaire, page 14, lines 13-17:

“A paper copy of the feedback questionnaire and pre-paid envelope will be posted to participants after the intervention appointment. This questionnaire contains 5-point Likert scale questions (e.g. strongly agree – strongly disagree) and free text questions about experiences of the checklist, appointment and action plan.”

- For the structured observations, it is noted that approx. 1 observation session for the recruitment/consent, intervention delivery and end of study clinical appointment will be conducted from each site – will there be multiple staff from each site performing each task? And therefore, more than 1 observation may be required? It would seem that observations of the intervention delivery component is quite important for fidelity – have the authors considered more observations of this aspect?

Thank you for this suggestion, which we have incorporated on page 15, lines 1-4.

“A target of three observations will be conducted for recruitment/ consent and end of study clinic appointments (one at each site). A target of two intervention appointments will be

observed per site (20%). More observations may be conducted if deemed necessary; for example, multiple clinical staff performing each procedure.”

- Authors state that 4-6 clinical staff are likely to be interviewed. Does this include both those involved in the recruitment/randomisation processes as well as the intervention delivery? As there is a mixture of clinical staff/intervention providers it is not 100% clear

Yes, clinical staff interviewed will be both those involved in the recruitment/randomisation processes and intervention delivery. We have clarified this on page 15, lines 15-16.

“The sample size is anticipated to be 8-10 patients and 4-6 clinical staff **(including those involved in recruitment/consent, intervention delivery and end of study clinic appointments).**”

- Additional information specifically related to the interviews should be included. For example, who will conduct/analyse these i.e a study staff member not involved in the study delivery? Are the interviews to be conducted face-to-face, or via phone, video? Will they be audio recorded, transcribed verbatim? Will an interview guide be used? What broad topics are to be covered?

Thank you for this suggestion, we have added further detail as follows:

Who will conduct the interviews, delivery and topic guide: page 15, lines 23-34

“Interviews will be conducted by GT, an experienced qualitative researcher. Interviews will be face-to-face (home/ hospital), telephone or video call, depending on the participants preference. Interviews will explore acceptability of the intervention and trial design. Semi-structured topic guides will include discussion of the following:

- Patient participants:
 - Intervention: intervention and trial design acceptability; how well intervention participants received and understood the intervention; extent to which intervention providers addressed needs; if the action plan was actioned; facilitators and barriers to enactment.
 - Control: trial design acceptability; intervention contamination.
 - Both: what care/support participants received; understanding what comprised usual care.
- Staff participants: acceptability of the trial design; experience of training day and understanding of the intervention; acceptability of delivering the intervention; facilitators and barriers to implementing both the trial design and the intervention; and experience of contamination with the control group.”

Interview audio recording and transcription, and who will analyse the interviews, page 15, lines 44:

“For qualitative data, **interviews will be audio recorded and transcribed verbatim**. NVivo v12 will be used to manage, sort, code and organise the anonymised transcribed data. Interview transcripts will be analysed **by GT** using directed thematic analysis informed by the research aims.”

Analysis:

- Is the planned analysis intention to treat?

Yes, we have added this to the analysis section, page 15, line 39-42:

“Analyses comparing the intervention and control groups will use the intention to treat principle, i.e. all participants will be analysed in the treatment group to which they were randomised irrespective of compliance or other protocol deviation.”

- Please define all ACRONYMS in tables/figures ie. eTable 1 PROMs used e.g. HADS, PAM-123

We have now defined all acronyms in the tables and figures – see appendix 2 and Figure 2 legend.

- What is the current trial status?

We have added trial status the methods section, page 5, lines 33-34:

“The study opened for recruitment in September 2021 with planned completion by December 2022.”

- Authors have included the SPIRIT guideline in the submission. If this has been used as a reporting guideline, then it would seem wise to refer to and cite in the protocol paper? Within the SPIRIT checklist, it is not clear what REC protocol refers to in relation to the submitted paper. Inclusion of these items in the paper is recommended ie. is there a process in place if participants want to withdraw, how are adverse events collected, what trial governance and data monitoring is in place?

We have added reference to the SPIRIT checklist on page 5, line 32:

“The study is reported in accordance with the SPIRIT checklist...”

Furthermore, we have added an additional appendix with further protocol information, including withdraw, adverse events, trial governance and data monitoring. See appendix 4.

VERSION 2 – REVIEW

REVIEWER	Crowfoot, Gary University of Newcastle, School of Health Sciences
REVIEW RETURNED	28-Apr-2022

GENERAL COMMENTS	Thank you for the opportunity to review the revision of this manuscript. I can appreciate the changes that have been made following my initial review. This protocol outlines an important study that will help to advance the field in this area. There are a couple of minor amendments that need to be addressed in this paper before publication: 1. There needs to be a clarification statement around who is blinded in the randomisation process. The intervention in the study does not allow for the clinicians delivering the intervention to be blinded but there should be a statement clearly outlining that the researchers are blinded to participant allocations.2. There needs to be clarity about who is revealing the arm the participants have been placed in and any impact this would have on the research team being blinded to the study. It reads as though it would be the clinicians but this has not been stated.3. The definitions used in the eligibility criteria should be cited to the authors that provided those definitions. For TIA this would be Easton's paper (https://pubmed.ncbi.nlm.nih.gov/19423857). Given
---

	the complexity and inconsistency around the definition of a minor stroke, the authors should clarify through citation if they have taken this particular definition from other published work.
REVIEWER	Purvis, Tara Monash University, Stroke and Ageing, School of Clinical Sciences at Monash Health
REVIEW RETURNED	29-Apr-2022
GENERAL COMMENTS	The authors should be congratulated as the additions and revisions made have improved the manuscript

VERSION 2 – AUTHOR RESPONSE

Thank you for the positive comments from reviewers on the revised manuscript.

Please see below a point-by-point response to address the final comments of

There are a couple of minor amendments that need to be addressed in this paper before publication:

1. There needs to be a clarification statement around who is blinded in the randomisation process. The intervention in the study does not allow for the clinicians delivering the intervention to be blinded but there should be a statement clearly outlining that the researchers are blinded to participant allocations.

We have added further information about blinding: Page 9, lines 26-27

“Due to the nature of the intervention, it is not possible to blind participants or clinicians delivering the intervention.”

Page 14, lines 1-3

“As this project is a training fellowship, the fellow (GT) will conduct the analysis and will have access to the whole dataset in order to conduct the trial. Therefore, it is not possible to conducted blinded analyses.”

2. There needs to be clarity about who is revealing the arm the participants have been placed in and any impact this would have on the research team being blinded to the study. It reads as though it would be the clinicians but this has not been stated.

We have added further detail about who is revealing the arm the participants: Page 9, lines 25-26:

“Participants will be notified of their allocation by a letter in the post, which will be sent by the research team at the Trials Unit.”

3. The definitions used in the eligibility criteria should be cited to the authors that provided those definitions. For TIA this would be Easton's paper (<https://pubmed.ncbi.nlm.nih.gov/19423857>). Given

the complexity and inconsistency around the definition of a minor stroke, the authors should clarify through citation if they have taken this particular definition from other published work.

Thank you for this suggestion, we have added the reference as requested. There is no standardised definition of minor stroke, therefore, we have added justification for the criteria we used (Box 1).